Corrected: Publisher correction

# Direct dioxygen evolution in collisions of carbon dioxide with surfaces

Yunxi Yao[1,2], Philip Shushkov[1,2], Thomas F. Miller III [1] & Konstantinos P. Giapis [1]

The intramolecular conversion of $CO_2$ to molecular oxygen is an exotic reaction, rarely observed even with extreme optical or electronic excitation means. Here we show that this reaction occurs readily when $CO_2$ ions scatter from solid surfaces in a two-step sequential collision process at hyperthermal incidence energies. The produced $O_2$ is preferentially ionized by charge transfer from the surface over the predominant atomic oxygen product, leading to direct detection of both $O_2^+$ and $O_2^-$. First-principles simulations of the collisional dynamics reveal that $O_2$ production proceeds via strongly-bent $CO_2$ configurations, without visiting other intermediates. Bent $CO_2$ provides dynamic access to the symmetric dissociation of $CO_2$ to $C+O_2$ with a calculated yield of 1 to 2% depending on molecular orientation. This unexpected collision-induced transformation of individual $CO_2$ molecules provides an accessible pathway for generating $O_2$ in astrophysical environments and may inspire plasma-driven electro- and photo-catalytic strategies for terrestrial $CO_2$ reduction.

---

[1] Division of Chemistry and Chemical Engineering, California Institute of Technology, Pasadena, CA 91125, USA. [2] These authors contributed equally: Yunxi Yao, Philip Shushkov. Correspondence and requests for materials should be addressed to K.P.G. (email: giapis@cheme.caltech.edu)

Althouth plentiful in modern Earth's atmosphere, molecular oxygen is extremely rare in space. Only trace amounts have been found elsewhere in our solar system[1–3] and in interstellar clouds[4,5]. The recent discovery of abundant $O_2$ in the coma of comet 67P/CG[6] has rekindled interest in abiotic reactions, occurring in extreme environments, which release $O_2$ from compounds, such as $H_2O$, $CO_2$, CO, silicates, and metal oxides. Such reactions may offer competing explanations for the origin of $O_2$ in comets, in the upper atmosphere of Mars, and in Earth's prebiotic atmosphere[7–9]. They may also present alternative ways for resource utilization related to space travel, such as generation of $O_2$ from $CO_2$ for making Mars habitable. Finally, new strategies for $CO_2$ activation may be inspired by such reactions.

The dissociation of $CO_2$ proceeds via multiple pathways depending on available energy. The partial dissociation reaction, $CO_2 \rightarrow CO + O$ ($^3P$ or $^1D$), has the lowest energy requirement (5.43 or 7.56 eV)[10]; it has been extensively studied in photochemistry and in heterogeneous catalysis under thermal activation conditions[11,12]. Full dissociation to $C + O + O$ involves the cleavage of both C–O bonds and requires 16.46 eV. Other pathways may be possible at intermediate energies, such as the exotic reaction: $CO_2 \rightarrow C(^3P) + O_2(^1\Sigma_g)$, which entails extensive intramolecular rearrangement of the $CO_2$ molecule. Calculations have suggested that this reaction proceeds on the ground-state potential energy surface, by first forming a cyclic $CO_2$ intermediate [c-$CO_2(^1A_1)$], which then rearranges into a collinear $COO(^1\Sigma^+)$ intermediate on its way to dissociation into $C + O_2$[10]. The first step in this channel involves bending of the $CO_2$ molecule to bring the two O atoms in close proximity, which requires close to 6 eV of internal energy[13].

Although inaccessible by thermal activation, transitions to electronically excited and anionic states of $CO_2$ can bend the molecule as a first step to $O_2$ production. Indeed, pioneering experiments employing VUV photo-excitation[14–16] and electron attachment[17,18] have shown that dissociation of $CO_2$ into $C(^3P) + O_2(X^3\Sigma_g^-)$ is possible, as evidenced by the detection of the complementary atomic $C^+$ or $C^-$ fragment. Further confirmation of the exotic pathway, however, remained elusive as neutral or ionized $O_2$ products were not detected. Using ion-beam scattering methods and numerical simulation techniques, we demonstrate here a different way to drive the direct reduction of $CO_2$ to $O_2$ with in situ detection of ionized $O_2$ products. The process involves a previously unknown intramolecular reaction pathway, which occurs in energetic $CO_2$ ion–surface collisions with a surprising lack of dependence on either the nature of the surface or the surface temperature. As such, the reaction may be relevant for astrophysical environments, such as comets, moons, and planets with $CO_2$ atmospheres.

## Results

### Carbon dioxide scattering experiments and kinematic analysis.

We first demonstrate the formation of $O_2$ in hyperthermal $CO_2^+/Au$ collisions by plotting kinetic energy distributions of three scattered molecular ion products: $CO_2^+$, $O_2^+$, and $O_2^-$ for various $CO_2^+$ incidence energies ($E_0$). Very weak signal of scattered $CO_2^+$ is detected for $E_0 < 80$ eV (Fig. 1a). The $CO_2^+$ exit energy peak varies in proportion to $E_0$, thus implying a ballistic or impulsive rebound from the surface and thereby precluding physical sputtering as its origin. Observation of this "dynamic" $CO_2^+$ signal is important, not only for proving that some $CO_2$ survives the surface encounter but also for unraveling the collision sequence of the constituent atoms. Strong signal of scattered $O_2$ ions is also observed (Fig.1b, c). The $O_2^+$ and $O_2^-$ exit energies represent a large fraction of the incidence energy (57%)

and increase monotonically with $E_0$ over a larger range than scattered $CO_2^+$. The $O_2$ ion signal intensity exhibits a maximum at $E_0 \sim 100$ eV. Above that, only the $O_2^+$ distribution develops a shoulder (i.e., exit at ~30 eV) from physical sputtering.

The detection of fast $O_2$ ion products is surprising. Neither sputtering of surface $O_2$ nor O-atom abstraction reactions (Eley–Rideal) can explain their formation, because both mechanisms would produce $O_2$ at much lower exit energies (see the section "Methods"). A remaining possibility to be explored here is dynamic formation of $O_2$ through dissociation of $CO_2$. Dynamic partial and full dissociation of $CO_2$ is in fact consistent with the other detected products, including $CO^+$, $CO^-$, $O^+$, $O^-$, and $C^+$ (Supplementary Fig. 1). The exit energy of the $CO^+$, $CO^-$, and $O^-$ fragments varies linearly with incidence energy, consistent with dynamic formation during the surface collision. In contrast, the $O^+$ and $C^+$ peaks show little dependence on $E_0$, suggesting a different origin, i.e., sputtering[19]. Scattered $C^+$ products appear at $E_0 > 80$ eV, confirming full dissociation.

The presence of dynamically exiting $CO_2^+$ ions enables use of kinematics[20] to clarify the scattering mechanism. Binary collision theory (BCT) allows calculation of the kinematic factor, defined as the fraction of incident energy retained by a scattered product exiting the surface. In the simplest possible model, $CO_2^+$ scatters as a whole molecule, i.e., a hard sphere with atomic mass of 44 Da. Under this assumption, BCT predicts a kinematic factor of 0.6349, which fits the data poorly (Fig. 2a) as may be expected given the quasi-linear nature of the triatomic $CO_2^+$ ion[21,22]. We consider next a kinematic model in which—as for diatomic molecules scattering on metal surfaces[23]—the leading O atom first collides with a surface Au atom, followed by a second collision of the CO moiety without prompt dissociation of the $CO_2$ molecule. Applying BCT to this sequential-collision model yields a kinematic factor of 0.7870, which agrees very well with the $CO_2^+$ exit energy data (Fig. 2a, black line).

The average exit energies for all remaining scattered products are also plotted in Fig. 2a. Potential origins for such species include partial or full dissociation of $CO_2$ and surface sputtering of adsorbed $CO_2$ fragments. While some sputtering is indeed observed at high $E_0$ (>140 eV), kinematic analysis of the exit energy data provides strong evidence for impulsive dissociation of the $CO_2$ molecule[24]. Assuming delayed fragmentation of the $CO_2$ parent[24], the kinematic factors of the CO, O, and (possibly) $O_2$ daughter products can be calculated from energy conservation to be 0.5724, 0.5008, and 0.2862, respectively. These factors are used as fixed slopes in one-parameter fittings of the respective data points (adjustable intercept). We find that the calculated slopes fit the $O_2^\pm$, $CO^\pm$, and $O^-$ ion exit data very well (Fig. 2a lines), indicating that the latter ions are all dissociation products of $CO_2$. On the contrary, the $O^+$ and $C^+$ data are not linear with respect to $E_0$, suggesting formation by other processes.

Velocity analysis for the observed scattered species provides further evidence regarding the collision mechanism. Figure 2b compares the ion distributions of various peaks for $E_0 = 56.4$ eV. The exit velocities of scattered $CO^+$, $O_2^+$, $O_2^-$, and the slower part of the $O^-$ distributions overlap, suggesting a common origin. However, the $O^-$ distribution is noticeably broader, extending to higher exit velocities, which suggests alternative formation channels. The $O_2$ ion products exit with velocities lower than $CO_2^+$ owing to inelasticity from breaking of chemical bonds and non-resonant surface ionization.

Although the kinematic analysis indicates conclusively that some $CO_2$ scatters intact after a two-step sequential collision of the O and CO moieties, it leaves various aspects of the $O_2$ formation mechanism unresolved. In particular, since the experiment is limited to observing ions, we are unable to assess how much neutral $O_2$ is produced. Moreover, the kinematic

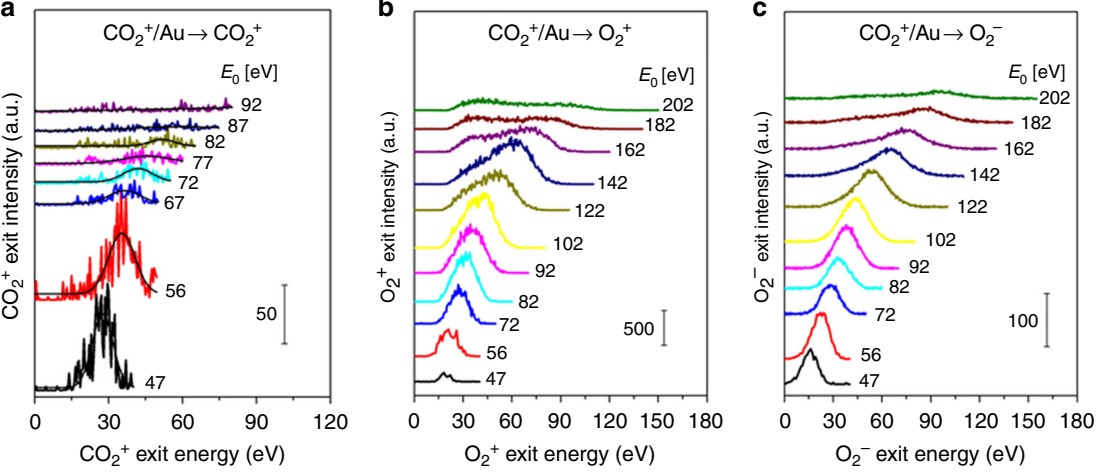

**Fig. 1** Dynamic production of $O_2^{\pm}$ in $CO_2^{+}$ collisions on Au. Scattered product kinetic energy distributions of **a** $CO_2^{+}$, **b** $O_2^{+}$, and **c** $O_2^{-}$ ion exits from $CO_2^{+}$/Au for various $CO_2^{+}$ beam energies ($E_0$) as annotated on each panel. Signal intensities in **b** and **c** cannot be compared to each other due to differences in detector bias

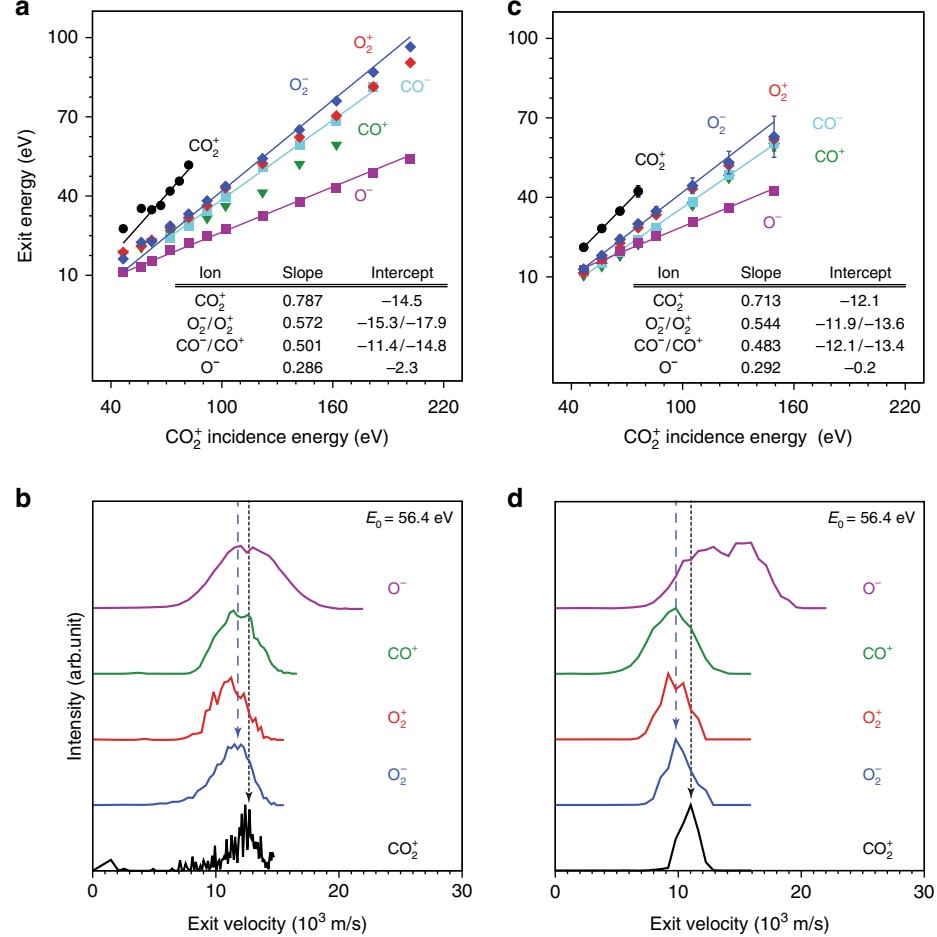

**Fig. 2** Kinematics and velocity analysis of $CO_2^{+}$ scattering on Au. **a** Experimental exit energies of $CO_2^{+}$, $O_2^{+}$, $O_2^{-}$, $CO^{+}$, $CO^{-}$, and $O^{-}$ ions from $CO_2$/Au collisions as a function of $CO_2^{+}$ incidence energy. All points represent the peak of the respective energy distribution obtained from Gaussian fitting of the experimental data. All solid lines represent one-parameter linear fittings with BCT-derived slopes. No fittings are shown for $O_2^{+}$ and $CO^{+}$ data because of overlap with their negative ion counterparts. **b** Experimental velocity distributions of select scattered product ions for $E_0 = 56.4 \pm 2.5$ eV. **c** Calculated exit energies of $CO_2^{+}$, $O_2^{+}$, $O_2^{-}$, $CO^{+}$, $CO^{-}$, and $O^{-}$ ions from MD simulations of $CO_2$/Au collisions as a function of $CO_2^{+}$ incidence energy; slopes and intercepts listed in the inset are two-parameter best-fittings. The error bars represent one standard deviation across 10 samples of 2000 trajectories each from the ensemble of molecular dynamics trajectories. **d** Calculated velocity distributions from MD simulations for select scattered ion products at $E_0 = 56.4$ eV. Vertical dashed lines in (**b**) and (**d**) indicate alignment with respect to the $CO_2^{+}$ (dashed black line) and $O_2^{-}$ (dashed blue line) peaks

analysis cannot shed light on whether $O_2$ is formed via an electronically adiabatic or non-adiabatic mechanism, nor can it disentangle the collision-induced pathways that underlie the exit velocity distributions of the ionic fragments. To address these questions, we next turn to first-principles molecular dynamics (MD) simulations.

**MD simulations of carbon dioxide collisions with gold.** MD trajectories for the scattering of $CO_2$ on Au(111) are performed in the experimental scattering geometry, with $CO_2$ evolving on the ground singlet potential energy surface under the assumption that incoming $CO_2^+$ ions are neutralized before the hard collision. Facile neutralization occurs via resonant electron tunneling[24–26] from the metal surface to the molecular cation because the molecular level of $CO_2$ (−13.8 eV) lies well within the occupied band of Au (−5.3 to −15.3 eV). Electron transfer from and to the surface is explicitly included in the simulations to also account for ionization of neutral collision products. The calculated exit energies of the products are plotted in Fig. 2c along with linear two-parameter fits. The slopes obtained from this fitting procedure compare very well to those determined from BCT (Fig. 2a). For example, the exiting $CO_2^+$ has a calculated slope of 0.713 vs. the experimental value of 0.787. Negligible $CO_2$ is found to survive for $E_0 > 80$ eV, consistent with the lack of experimental signal at these energies. All other calculated slopes agree well with the experimental values; for instance, compare the slope of $0.54 \pm 0.02$ vs. the experimental value of 0.57 for the $O_2^-$ line. These results indicate broad agreement between the simulations and the scattering kinematics.

The formation of ions detected in the experiment requires surface ionization, which influences the yields of the ionic products. The MD simulations demonstrate a substantial enrichment of $O_2^-$ ions over $O^-$, resulting from the exponential dependence of the ionization probability on the coupling to the metal surface (Supplementary Fig. 2, red curve), which can reach ~30%, comparable to the experimentally derived yield of 33% (Supplementary Fig. 2, blue curve).

The agreement between experiment and simulations is further demonstrated by comparing the ion exit velocity distributions at $E_0 = 56.4$ eV (Fig. 2d). Although the experimental peak positions appear systematically at somewhat larger velocities than the calculated ones, the distributions agree very well with respect to relative position of the peaks. In particular, both simulations and experiment find the $CO^+$ and $O^-$ velocity distributions to be broadened, both find the $O_2^+$ and $O_2^-$ distributions to be similar with the cation exiting slower than the anion, and both find $CO_2^+$ to exit with higher velocity than the ionized $O_2$ products. The agreement suggests that the simulations provide a strong foundation for analyzing the reaction mechanism of the direct $CO_2$ conversion to $O_2$.

An ensemble of 20,000 $CO_2$-on-Au collision trajectories were performed for each incidence energy, resulting in a variety of dissociation products, including $O_2$ (Fig. 3a). Prior to the mechanistic ensemble analysis, it is instructive to review one representative trajectory that leads to collisional $O_2$ formation (Fig. 3b). Select configurations are shown as insets, along with the $CO_2$-Au interaction energy, $E_{CO_2-Au}$, and the $CO_2$ internal energy, $E_{CO_2}$, as a function of time. The incoming $CO_2$ molecule is vibrationally excited (inset I). As the center-of-mass distance to the surface, $Z_{CO_2}$, decreases, the molecule penetrates the repulsive potential wall of the surface and $E_{CO_2-Au}$ increases steeply. During this encounter, one of the O atoms of $CO_2$ strikes a surface Au atom, giving rise to the first peak in the $E_{CO_2-Au}$ curve (inset II). This collision occurs before $Z_{CO_2}$ reaches a minimum at the apsis

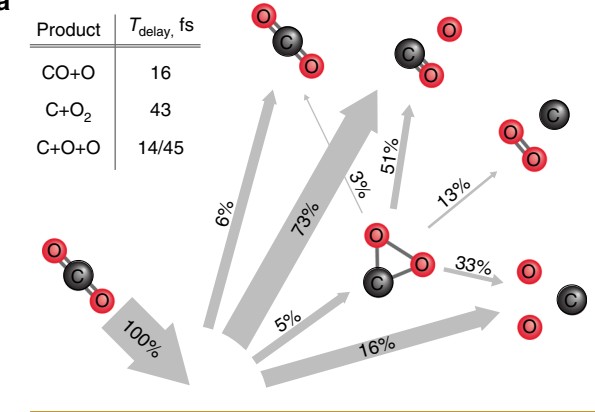

| Product | $T_{delay}$, fs |
|---------|-----------------|
| CO+O    | 16              |
| C+$O_2$ | 43              |
| C+O+O   | 14/45           |

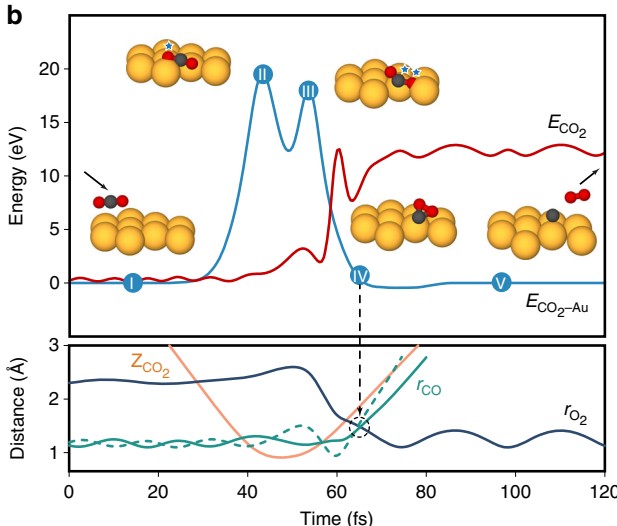

**Fig. 3** Product yields and energetics of $CO_2$ scattering on Au. **a** Calculated yields of neutral dissociation products from a statistical analysis of an ensemble of $CO_2$ scattering trajectories on Au(111) at $E_0 = 56.4$ eV. Inset: average delay times ($T_{delay}$) of the dissociation channels relative to the point of closest approach of $CO_2$ to the surface. **b** Energetics along an illustrative $CO_2$ collision trajectory, which leads to $O_2$ formation on Au at $E_0 = 56.4$ eV. Curve identification: $CO_2$–Au interaction energy (light blue), $CO_2$ potential energy (red), $CO_2$ center-of-mass distance from the metal surface (orange), O–O bond length (dark blue), and C–O bond lengths (full and dashed green lines). Insets I–V: Select configurations of the $CO_2$ surface geometry along the scattering trajectory

point. As the O atom rebounds, the CO moiety collides with a different Au atom, causing a second peak in the $E_{CO_2-Au}$ curve (inset III), which occurs after the apsis. As a result of the impulsive energy transfer during the collision, the rebounding $CO_2$ undergoes substantial intramolecular rearrangement portrayed by the bond distance evolution in Fig. 3b. The O–O distance, $r_{O_2}$, decreases while the C–O distances, $r_{CO}$, simultaneously increase, reaching a point along the trajectory where $CO_2$ acquires a triangular configuration with nearly equal bond lengths (vertical dashed line). This strongly bent $CO_2$ intermediate (inset IV) has a significant amount of internal energy, $E_{CO_2}$, and promptly dissociates to give a free C atom and a vibrationally hot $O_2$ molecule (inset V). The complete $CO_2$ collision trajectory discussed in Fig. 3b can be viewed in the Supplementary Video. The formation of $O_2$ depicted by this representative trajectory proceeds by delayed fragmentation following the two-step

sequential collision of $CO_2$ with the surface. This mechanism is consistent with the assumptions of the kinematic model used earlier to explain the experimental data in Fig. 2a, b.

The calculated reaction yields of the various collision-induced dissociation channels of $CO_2$ at $E_0 = 56.4$ eV are shown in Fig. 3a. As expected for this low incidence energy, the partial dissociation channel dominates the reaction yield with 73% of all MD trajectories taking that pathway. The complete dissociation channel is second at 16%. A small fraction of the incoming $CO_2$ (6%) survives the collision in correspondence with experimental detection. Approximately 5% of all trajectories lead to the strongly bent intermediate state—the precursor to $O_2$ formation—which is characterized by C–O and O–O bond orders exceeding 0.7. This intermediate state fragments primarily via partial dissociation (51%) followed again by complete dissociation, albeit now with a higher yield (33%). Remarkably, one in eight (13%) of the strongly bent $CO_2$ molecules produces $O_2$. The overall neutral yield of the symmetric dissociation channel, $CO_2 \rightarrow C + O_2$, amounts to 0.6% at $E_0 = 56.4$ eV. Upon increasing incidence energy, the neutral $O_2$ yield obtained from the ensemble of isotropically oriented incident $CO_2$ molecules reaches $0.8 \pm 0.2\%$ for $E_0 \sim 70 \pm 15$ eV (Fig. 4, blue line). Also it is clear from the figure that the fraction of $O_2$-producing trajectories increases substantially once the strongly bent $CO_2$ intermediate state is reached (Fig. 4, green line) and this fraction peaks at around 13% for $E_0 \sim 55 \pm 10$ eV. The smaller total neutral $O_2$ yield results from the small fraction of linear $CO_2$ molecules reaching the strongly bent state (Fig. 4, red line). By preferentially orienting incoming $CO_2$ molecules (axis parallel to the surface), the fraction of $O_2$-producing trajectories increases to ~2% (Fig. 4, dashed blue line) resulting from an increase of the dissociation probability of the strongly bent state to $O_2$ (Fig. 4, dashed green line). These findings suggest that activation of bending and symmetric stretching motion of $CO_2$ prior to the collision may facilitate both the population of the strongly bent state and its

dissociation to $O_2$ leading to a significant increase in the total neutral $O_2$ yield.

The timescales for bond breaking and formation in the collision-induced dissociation reactions were evaluated for $E_0 = 56.4$ eV and are reported in the inset of Fig. 3a. The average delay times reveal that both partial dissociation and the first C–O bond-breaking event in complete dissociation occur promptly after the apsis. In contrast, the formation of the strongly bent $CO_2$ intermediate state and its fragmentation to $O_2$ occur on a longer timescale, to allow for the significant intramolecular rearrangement that precedes symmetric dissociation. This is again consistent with the assumption of delayed fragmentation used in the kinematic modeling. The second C–O bond breaking in the complete dissociation channel is also delayed, irrespective of the degree of bending of $CO_2$. The different timescales of the collisional reactions explain the widths of the observed exit velocity distributions. For instance, O atoms produced in prompt partial and delayed complete dissociation, have different velocity profiles, giving rise to a considerably broader $O^-$ velocity distribution (Fig. 2d). In particular, prompt partial dissociation involves direct scattering of O atoms from the much heavier Au target, producing faster O-atom exits owing to inefficient momentum transfer. On the other hand, the second C–O bond breaking involves dissociation of the more massive, recoiling CO moiety of $CO_2$, which gives off slower O atoms (Supplementary Fig. 3). Moreover, the narrow velocity profiles of the molecular $O_2$ ions stem from $CO_2$ scattering as a whole molecule, which breaks apart unimolecularly during the rebound from the surface.

## Discussion

The convergent analysis and agreement among experiment, kinematics, and first-principles MD simulations presented in this work support a collision-induced mechanism for direct intramolecular conversion of $CO_2$ to $O_2$. Specifically, with the dynamics evolving on the ground electronic state of neutral $CO_2$, we find that $O_2$ is formed via delayed fragmentation, where the delay results from atomic rearrangement of the colliding $CO_2$ molecules into a strongly bent geometry. This geometry provides access to the $O_2$ dissociation product, without visiting other intermediates. Alternative mechanisms were also theoretically investigated, including the possibility of a collision-induced, non-adiabatic transition of the neutral $CO_2$ molecules to electronically excited states (Supplementary Fig. 4), as well as collisional dissociation on the anionic $CO_2^-$ surface following double electron transfer from the Au surface. Although these more complicated processes offer intriguing and potentially exploitable alternative avenues to $O_2$ formation, they were not necessary for explaining the experimental observables and were calculated to be less likely under the current experimental conditions.

The mechanism reported here is distinct from previously proposed mechanisms for $CO_2 \rightarrow C + O_2$ conversion. Specifically, the mechanism differs from that of photochemical interconversion[14] not only in terms of activation (collisional vs. photochemical) but also because the collisional mechanism occurs via a delayed fragmentation of a single $CO_2$ intermediate, i.e., without visiting the linear COO state. The collisional mechanism also differs fundamentally from that taking place in electron-attachment experiments[17], where the $CO_2$ bends spontaneously on the anionic potential energy surface. Instead, the bent $CO_2$ state is accessed on the neutral surface via collisional energy transfer. Furthermore, while the collisional interconversion of $CO_2$ to $O_2$ has comparable efficiency to activation via high-energy photons and higher efficiency than via electron attachment, it is a much simpler process. Importantly, our mechanism is independent of surface temperature and generic to

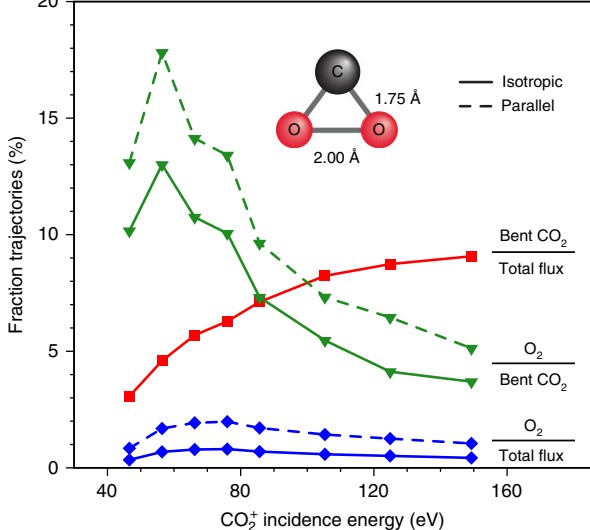

**Fig. 4** Reaction probabilities for symmetric dissociation of $CO_2$ on Au. Calculated total $O_2$ yields for isotropic (solid blue line) and parallel (dashed blue line) orientation of incident $CO_2$ as a function of incidence energy. Also shown: Fraction of all trajectories reaching the strongly bent $CO_2$ intermediate state (solid red line); and fraction of the latter trajectories, which dissociate symmetrically to $C + O_2$ for isotropic (solid green line) and parallel (dashed green line) orientations. Inset: average geometry of the intermediate state computed from the lowest energy, bent $CO_2$ configurations visited by the scattering trajectories

surface composition (tested on Au, Pt, $SiO_2$, $In_2O_3$, $SnO_2$) as long as: (i) the surface contains atoms heavier than the constituents of $CO_2$ and (ii) surface charging is mitigated when $CO_2$ ions are used. Finally, we note that an analogous dissociation reaction: $OCS \rightarrow C + SO$, previously reported[27] for $OCS^+$ collisions on Ag (111), was speculated to occur via a sharply bent excited state, such as the $OCS(^3A)$, activated either by neutralization prior to impact or by the energetic collision with the surface. However, the basic mechanistic features of the latter process—including whether it involves unimolecular collisions or Eley–Rideal reactions with surfaced-absorbed O or S atoms—were not addressed.

The intramolecular $CO_2$ reaction may be relevant in astrochemical environments with abundant $CO_2$ and prevalent solar wind. Solar ultraviolet light photo-ionizes $CO_2$ molecules readily, producing ions which are then picked up by the solar wind and accelerated to hyperthermal energies[28,29]. Collisions of these fast ions with the surfaces of dust particles or other astrophysical bodies can activate the dissociation. Such interactions may affect dynamically the composition of cometary comae, contributing to the abundance of the super-volatiles $O_2$ and CO. Production of $O_2$ from $CO_2$ was explicitly disregarded in the coma of comet 67P early on (pre-perihelion) during the Rosetta mission, owing to the low abundance of $CO_2$ and poor correlation between $O_2$ and $CO_2$ fluxes[6]. However, the situation may warrant reexamination in the post-perihelion phase, when $CO_2$ can reach abundancies as high as 32% relative to $H_2O$, a 10-fold increase versus pre-perihelion[30]. The precise level of contribution to the $O_2$ abundance in the coma cannot be determined without $CO_2$ ion energy and flux data. Nevertheless, the number is likely small for collisional encounters on dust and cometary surfaces. Even at low yield, however, contribution to the measured $O_2$ abundance may be disproportionate if the $CO_2$ reaction occurs close to the point of measurement. For example, we have verified experimentally that the reaction takes place on indium–tin oxide (ITO), a man-made material found on Rosetta's thermal insulation and solar panels. Thus, $CO_2$ collisions on the spacecraft's exposed surfaces can change the composition of the surrounding gaseous halo with unknown repercussions for mass spectrometric measurements[31].

Similar collisional processes may have occurred in early Earth when projectiles, such as meteorites, traversed through its $CO_2$-dominated atmosphere; likewise, orbiting satellites/spacecraft or high-speed space debris[32] will encounter neutral or photoionized $CO_2$ in Mars' upper ionosphere. In these situations, the target surface is moving against a stagnant $CO_2$ atmosphere with correspondingly high velocities, driving the partial transformation of $CO_2$ into $O_2$. Indeed, $O_2$ abundances in the 1000's of parts per million measured at Mars[33] may contain significant contributions from such processes.

Finally, although the yield of $O_2$ is relatively small in the current study, a combination of collisional activation with photoexcitation, electron attachment, and Eley–Rideal reactions in a plasma reactor may result in a process that could be promising for $CO_2$ reduction strategies, as well as plasma-driven continuous $O_2$ production in $CO_2$ atmospheres.

## Methods

**Experimental**. All experiments were carried out in a custom-made low-energy ion scattering apparatus[34]. The $CO_2^+$ ion beam was extracted from an inductively coupled plasma, struck in a reactor held at 2 mTorr using a $CO_2$/Ar/Ne gas mixture supplied with 500 W RF power at 13.56 MHz. Ions were delivered to a grounded surface at 45° incidence angle; typical beam currents of 5–15 μA were spread over a ~3 mm spot. Beam energy was varied between 40 and 200 eV by externally adjusting the plasma potential. The beam energy distribution had a Gaussian shape with a FWHM of ~5 eV. Typical target surfaces were polycrystalline Au foils (5 N), sputter-cleaned with an $Ar^+$ ion gun before each run. Scattered ion products, exiting at an angle of 45° in the scattering plane, were energy-resolved and mass-resolved using an electrostatic ion energy analyzer and a quadruple mass spectrometer, respectively. All ions were detected using a channel electron multiplier,

biased as appropriate to detect positive or negative ions. Differences in detector bias precluded a direct comparison of signal intensities between product ions of different charge polarities. All collected signals were normalized to the beam current measured on the sample.

**Simulations**. MD trajectories of $CO_2$ scattering from Au(111) surface were propagated on a potential energy surface represented as a sum of the $CO_2$ electronic energy, the molecule–surface interaction energy, and the interatomic potential of the metal atoms. The $CO_2$ energy of the ground singlet electronic state was modeled by a self-consistent-charge tight-binding model, the molecule–surface interaction energy was described by a Tersoff–Brenner reactive force field, and the interatomic metal interactions were represented by an embedded-atom potential. Both the tight-binding model and reactive force field were specifically parameterized for the $CO_2$–Au system on datasets consisting, respectively, of 689 and 930 ab initio energies with mean absolute errors of 0.42 and 0.48 eV, which are small compared to the translational and internal energies involved in the collisions. Finite-temperature tight-binding calculations were used to alleviate concerns about the multi-reference nature of the dissociating scattering products. Non-adiabatic transition probabilities presented in Supplementary Fig. 4 were computed by integration of the time-dependent Schrödinger equation, together with on-the-fly multi-reference complete active space self-consistent field calculations of the participating ground and excited electronic states. The Langevin equation with a friction coefficient that described the dissipation of energy in electron–hole pair excitations in the metal was integrated to propagate the motion of the atoms. The Au(111) surface was modeled by a $(8 \times 8 \times 6)$ slab of Au atoms with 2D periodic boundary conditions imposed in the $x$ and $y$ directions. Trajectories were initiated with fixed incidence angle of 45° and with center-of-mass of the $CO_2$ molecules at 5.5 Å above an energy-minimized gold surface, and they were terminated after the scattering products had left the surface region. The trajectories were averaged over molecular orientation and surface unit cell, and 0.2 eV thermal ro-vibrational energy was attributed to the internal degrees of freedom of $CO_2$. 20,000 trajectories were computed for each incidence energy and all scattering products were collected for final analysis. The probability for ion formation depended exponentially on the inverse of the normal component of the center-of-mass velocity and the coupling to the metal, and the exit velocities of the ions were corrected for the ionization energies.

**Competing mechanisms**. Before considering an intramolecular reaction, two competing processes must be excluded: (1) sputtering of $O_2$ from the surface and (2) abstraction of atomic O from the surface to form the hypothetical radical $\cdot CO_3$ via an Eley–Rideal reaction, followed by spontaneous dissociation to $CO + O_2$. Sputtering can be discounted for two reasons: (a) sputtering requires typically higher $E_0$ to induce the collision cascade—indeed, there is some evidence of sputtering in the $O_2^+$ energy distributions for $E_0 > 100$ eV, forming a low-energy shoulder to the main peak (Fig. 1b) and (b) the sputtering peak position varies little with $E_0$—indeed, the main $O_2^+$ peak separates from the low-energy shoulder entirely for $E_0 > 180$ eV (Fig. 1b). Moreover, there is no evidence of sputtering in the $O_2^-$ energy distributions (Fig. 1c). Similarly, formation of $O_2$ by Eley–Rideal reactions likely does not occur, owing to low sticking probability for O on Au. In addition, abstraction reactions slow down the exiting product moiety and would result in $O_2^\pm$ exit energies much lower than those measured[20]. Isotopic scattering experiments with $C^{16}O_2^+$ on $^{18}$O-covered Pt surfaces have confirmed that an Eley–Rideal reaction is possible and that it yields slower $^{16}O^{18}O^+$ than the simultaneously observed $^{16}O^{16}O^+$ products (Supplementary Figs. 5 and 6). The energy peaks from $CO_2^+$ scattering on Au show no such contribution at lower energies—the peaks are much narrower. Therefore, neither $O_2$ sputtering nor O-atom abstraction by $CO_2$ on Au surfaces occurs to a degree that would affect the conclusions of our study.

## Data availability

All relevant raw data, experimental and computational, are available from the authors upon request.

## Code availability

The computational code is available from the authors upon request.

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

## Acknowledgements

This report was based on work funded by NSF (Award no. 1202567) and by the Joint Center for Artificial Photosynthesis, a DOE Energy Innovation Hub, supported through the Office of Science of the U.S. Department of Energy (Award no. DE-SC0004993). P.S. is grateful for a postdoctoral fellowship funded by the Deutsche Forschungsgemeinschaft.

## Author contributions

Y.Y. and K.P.G. designed the experiments. P.S. and T.F.M. designed the simulations. Y.Y. conducted experimental measurements, while P.S. performed the computations. All authors participated in analyzing the results and writing the paper.

## Additional information

**Competing interests:** The authors declare no competing interests.

