## [Peer Review File · Nature Communications]

Reviewers' comments:

Reviewer #1 (Remarks to the Author):

This report introduces an unusual mechanism for fragmentation of the CO₂ molecule. Rather than breaking a C-O bond, which is the typical and lowest energy pathway, the authors propose a pathway where the linear molecule bends (a pathway with a high barrier) and pops out the central carbon atom to form the O₂ molecule. This unusual chemistry is of considerable interest from a fundamental chemical dynamics perspective (and thus of significant interest to the chemical physics community). But as the authors point, their result could have implications in trying to understand the occurrence of molecular oxygen in the absence of life (whose chemical signature is the conversion of carbon dioxide into oxygen) or in interstellar clouds.

This is not the first study to propose formation of molecular oxygen from the CO₂ molecule - dissociative electron attachment and vacuum UV photodissociation pathways have been proposed before, both in high profile papers in Science and Nature Chemistry. What is different in the current paper is that the molecular oxygen fragment is detected directly. While detection of the carbon atom fragment can provide just as hard evidence (and the authors should not dismiss prior studies as "complicated experimental means of CO₂ activation" - they are standard chemical physics methods) there is value in capturing the molecular oxygen fragment and utilizing a very different experimental methodology. I would like to see the authors spend a more appropriate quantity of description in the Introduction explaining the previous work properly, and what they measured and how they contrast with the current work.

The authors use high energy collisions (much above typical conditions found at ordinary temperatures) to seek signatures of this unusual pathway. A core assumption in this paper, and previous papers by this and other groups, is that using a positive ion (in this case CO₂⁺) which can much more easily be accelerated and slammed into a metal surface, is neutralized by an electron transfer event (harpooning) from the metal just as the incoming projectile gets close to the surface. From that point on, the system and the trajectory it follows, is described as belonging to neutral CO₂. Although the authors provide a lead reference on this point, it would be helpful to explain this assumption, and the evidence for it, much more carefully if the intention is to reach a broader readership.

Otherwise the paper is well constructed and argued. The experimental data, particularly the observation of negative ions such as O₂⁻, support the author's initial assumption about the metal electrons participating in the molecule/surface encounter. Appropriate control experiments are provided.

The proposed mechanism is supported by high quality molecular dynamics calculations that help to visualize the atomistic mechanism. Reassuringly, these calculations include the electron transfer aspects of the metal surface behavior with necessary non-adiabatic transitions computed on the fly. The calculations provide predictions for the velocity distributions in the emerging products, which agree relatively well with experiment. And the wide variety of metal surfaces (and more surprisingly metal oxide surfaces) tried in the experiment, suggests the generality of the newly discovered pathway. Why do the authors think that metal oxide surfaces can harpoon an incoming CO₂⁺?

I find the paper suitable for publication in Nature Comm as it is of significant value to the disciplinary area and also has broader impact outside the direct chemical physics community. The authors should however take on board the two recommendations included in the above text.

Reviewer #2 (Remarks to the Author):

My comments pertain exclusively to the motivation and impact statements in the paper and I leave it to other referees with more expertise in the proposed chemistry to review the experimental results. The authors appear to provide four motivations for the study:

1. Explaining the molecular oxygen in the coma of 67P
2. The impact of the reaction of Earth's early prebiotic atmosphere
3. The habitability of exoplanets for Earth-like life
4. O₂ production on Mars

None of these seem like good fits with the discovered O₂ production pathway from CO₂ collision as presented in the paper, which also limits the impact of the results unless there are other environments where this reaction may play a larger role. I explain in more detail below.

1. It seems like what the authors are proposing is that the observed O₂ does not originate from outgassing from 67P, but rather from chemistry in the coma. While it is not completely clear in the paper, the authors seem to imagine the Solar wind accelerating outgassing CO₂ to 10s of eV and then convert some of that CO₂ into O₂. While it is not impossible that some of the outgassing CO₂ is indeed accelerated to the required velocities, the paper would have benefitted from some order-of-magnitude calculations to show what fraction of the outgassing CO₂ would be affected. More seriously, the yield of the O₂ production yield of the accelerated CO₂ is only ~1% according the experiments. This implies that we should observe a O₂/CO₂ fraction in the coma that is <1% taking

into account both the yield and the fact that not all outgassing CO₂ is thus accelerated. The measured ratio is however close to unity.

2. Based on the statements in the paper and the references it is unclear whether the authors are proposing that comet-produced O₂ is delivered to the young Earth or O₂ produced through CO₂ collisions in Earth's atmosphere. The former does not seem a good fit with the assumptions in 1., i.e. that the O₂ in comets are not indigenous to the comet, but rather produced through collisions in the comet coma. And even if some of the coma-produced O₂ re-adsorbed on the comet, the O₂ delivered to Earth this way would have a rather short lifetime in Earth's atmosphere. The second interpretation is also difficult to understand, since the authors do not give any source for fast CO₂ collisions on the young Earth.

3. Similar to the Earth-case, it is unclear whether comet-impact delivered oxygen would survive long enough to make a difference. Even transient oxygenation events may be important for remote atmosphere evaluations, but as mentioned above the proposed reaction would not in fact affect comet bulk O₂. If the authors are instead imagining local production channels through accelerated CO₂ collisions, this needs to be made explicit and motivated.

4. It is difficult to imagine that accelerating CO₂ to 10s of eV would be a competitive O₂ production pathway on Mars. This motivation would seem more realistic if there are reasons to think that this conversion chemistry could take place in less energetic impacts than used in this experiment.

Reviewers' comments:

Reviewer #1 (Remarks to the Author):

This report introduces an unusual mechanism for fragmentation of the CO₂ molecule. Rather than breaking a C-O bond, which is the typical and lowest energy pathway, the authors propose a pathway where the linear molecule bends (a pathway with a high barrier) and pops out the central carbon atom to form the O₂ molecule. This unusual chemistry is of considerable interest from a fundamental chemical dynamics perspective (and thus of significant interest to the chemical physics community). But as the authors point, their result could have implications in trying to understand the occurrence of molecular oxygen in the absence of life (whose chemical signature is the conversion of carbon dioxide into oxygen) or in interstellar clouds.

Answer: We thank the reviewer for the praise and for noting the broader implications of our work.

This is not the first study to propose formation of molecular oxygen from the CO₂ molecule - dissociative electron attachment and vacuum UV photodissociation pathways have been proposed before, both in high profile papers in Science and Nature Chemistry. What is different in the current paper is that the molecular oxygen fragment is detected directly. While detection of the carbon atom fragment can provide just as hard evidence (and the authors should not dismiss prior studies as "complicated experimental means of CO₂ activation" - they are standard chemical physics methods) there is value in capturing the molecular oxygen fragment and utilizing a very different experimental methodology. I would like to see the authors spend a more appropriate quantity of description in the Introduction explaining the previous work properly, and what they measured and how they contrast with the current work.

Answer: We appreciate the reviewer's noting of the important difference of our work from that presented in the cited high profile papers.

We revised the text to remove the offending statement about prior studies and to give more credit where credit is due. We still believe that detection of O₂ is key to the mechanism as it removes any and all ambiguity pertaining to its formation. The revised paragraph is copied below:

“Although inaccessible by thermal activation, transitions to electronically excited and anionic states of CO₂ can bend the molecule as a first step to O₂ production. Indeed, pioneering experiments employing VUV photo-excitationⁱ⁻ⁱⁱⁱ and electron attachment^{iv,v} has shown that dissociation of CO₂ into C(³P) + O₂(X³Σ_g⁻) is possible, as evidenced by the detection of the complementary atomic C⁺ or C⁻ fragment. Further confirmation of the exotic pathway, however, remained elusive as neutral or ionized O₂ products were not detected. We report here an entirely different way to drive the direct reduction of CO₂ to O₂ with in-situ detection of ionized O₂ products. The process involves a previously unknown intra-molecular reaction pathway, which occurs in energetic CO₂ ion-surface collisions though, surprisingly, it depends on neither the nature of the surface nor the surface temperature. As such, the reaction may be relevant for astrophysical environments such as comets, moons, and planets with CO₂ atmospheres.”

The authors use high energy collisions (much above typical conditions found at ordinary temperatures) to seek signatures of this unusual pathway. A core assumption in this paper, and

previous papers by this and other groups, is that using a positive ion (in this case CO_2^+) which can much more easily be accelerated and slammed into a metal surface, is neutralized by an electron transfer event (harpooning) from the metal just as the incoming projectile gets close to the surface. From that point on, the system and the trajectory it follows, is described as belonging to neutral CO_2 . Although the authors provide a lead reference on this point, it would be helpful to explain this assumption, and the evidence for it, much more carefully if the intention is to reach a broader readership.

Answer: Neutralization of CO_2^+ at the metal surface proceeds via resonant tunneling of an electron from the metal surface to the molecular cation. The resonant charge exchange mechanism is available because the molecular level of CO_2 (-13.78 eV) lies well within the occupied band of Au (-5.3 to -15.3) and is very efficient as demonstrated by computed neutralization rates [N. Lorente, D. Teilette-Billy, J.-P. Gauyacq, Surf. Science 402, 197 (1998)]. Simulations show that all incoming molecular cations are efficiently neutralized prior to the collision with the surface consistent with a number of experimental measurements [N. Lorente, D. Teilette-Billy, J.-P. Gauyacq, Nucl. Instrum. Methods B 157, 1 (1999), D. C. Jacobs, Ann. Rev. Phys. Chem. 53, 379 (2002) and references therein]. Similar conclusions also hold for semiconductor surfaces, such as metal oxides.

We have added a shortened version of this explanation in the text, as follows:

Facile neutralization occurs via resonant electron tunneling new_refs from the metal surface to the molecular cation because the molecular level of CO_2 (-13.78 eV) lies well within the occupied band of Au (-5.3 to -15.3 eV).

Otherwise the paper is well constructed and argued. The experimental data, particularly the observation of negative ions such as O_2^- , support the author's initial assumption about the metal electrons participating in the molecule/surface encounter. Appropriate control experiments are provided.

The proposed mechanism is supported by high quality molecular dynamics calculations that help to visualize the atomistic mechanism. Reassuringly, these calculations include the electron transfer aspects of the metal surface behavior with necessary non-adiabatic transitions computed on the fly. The calculations provide predictions for the velocity distributions in the emerging products, which agree relatively well with experiment. And the wide variety of metal surfaces (and more surprisingly metal oxide surfaces) tried in the experiment, suggests the generality of the newly discovered pathway. Why do the authors think that metal oxide surfaces can harpoon an incoming CO_2^+ ?

Answer: The reviewer points out correctly to a surface charging problem relating to ion interactions with metal oxide surfaces. While conductive metal oxides (e.g., Indium Tin Oxide, ITO) pose

obviously no problem, insulating metal oxides could charge up and, then, influence the harpooning process. Our experiments on such oxides (SiO_x, FeO_y) take place on very thin layers formed in-situ on conducting substrates (e.g., SiO₂ on heavily doped Si) so that electron tunneling to or from the substrate can still take place. That way, the harpooning mechanism is not impeded. We have added a sentence to disclose this important experimental fact:

Importantly, it is independent of surface temperature and generic to surface composition (tested on Au, Pt, SiO₂, In₂O₃, SnO₂) as long as: 1) a collision partner atom exists on the surface which is heavier than the constituents of CO₂, and 2) there is a path for electrons from the surface to the underlying conductive substrate to mitigate surface charging.

I find the paper suitable for publication in Nature Comm as it is of significant value to the disciplinary area and also has broader impact outside the direct chemical physics community. The authors should however take on board the two recommendations included in the above text.

Answer: We appreciate the vote of confidence and the time spent for thorough review.

Reviewer #2 (Remarks to the Author):

My comments pertain exclusively to the motivation and impact statements in the paper and I leave it to other referees with more expertise in the proposed chemistry to review the experimental results. The authors appear to provide four motivations for the study:

1. Explaining the molecular oxygen in the coma of 67P
2. The impact of the reaction of Earth's early prebiotic atmosphere
3. The habitability of exoplanets for Earth-like life
4. O₂ production on Mars

None of these seem like good fits with the discovered O₂ production pathway from CO₂ collision as presented in the paper, which also limits the impact of the results unless there are other environments where this reaction may play a larger role. I explain in more detail below.

Answer: We appreciate the reviewer's expounding on the relevance of our proposed chemistry to astrophysics, in fact, we were very much hoping our work would motivate readers to think along the presented lines. Motivations for the study are just that: we are trying to convey succinctly the broader applicability of the new reaction to give readers, in particular astrophysicists, points of departure. The fit is better for some motivations and worse for others, however, we were careful to avoid exaggeration. We address the reviewer's 4 points explicitly below, and while we could expand on the reasons we believe the work is relevant, we feel it will lengthen the introduction unnecessarily.

1. It seems like what the authors are proposing is that the observed O₂ does not originate from outgassing from 67P, but rather from chemistry in the coma. While it is not completely clear in the paper, the authors seem to imagine the Solar wind accelerating outgassing CO₂ to 10s of eV and then convert some of that CO₂ into O₂. While it is not impossible that some of the outgassing CO₂ is indeed accelerated to the required velocities, the paper would have benefitted from some order-of-magnitude calculations to show what fraction of the outgassing CO₂ would be affected. More seriously, the yield of the O₂ production yield of the accelerated CO₂ is only ~1% according to the experiments. This implies that we should observe a O₂/CO₂ fraction in the coma that is <1% taking into account both the yield and the fact that not all outgassing CO₂ is thus accelerated. The

measured ratio is however close to unity.

Answer: We do *not* propose “that the observed O₂ does not originate from outgassing from 67P”. We have been very careful in our wording to point out that the new CO₂→O₂ mechanism may only *contribute* to the observed O₂ abundance. Indeed, CO₂ ions have been discovered in the extended 67P coma, formed by photo-ionization. 67P is considered to be a CO₂-rich comet [Bockelee-Morvan et al. MNRAS 462, S170 (2016)], and CO₂ can reach up to 32% relative to H₂O in the coma post-perihelion, a fairly high abundance. Like water ions, CO₂ ions can be picked up by the solar wind and then reach energies in the 100’s eV, similar to the ones we studied. Such collisions, whether on cometary surfaces (silicates, iron oxides) or on Rosetta surfaces (indium tin oxide), will produce some O₂ thus *contributing* to the observed O₂. We modified the text to reflect this discussion:

“This reaction may be producing O₂ in astrochemical environments with abundant CO₂ and prevalent solar wind. Solar UV-light can photo-ionize CO₂ readily, then the ions can be picked up by the solar wind and accelerated to hyperthermal energies,^{vi,vii} before undergoing collisions with solid surfaces of dust particles or other astrophysical bodies. Such energetic collisions may influence the composition of cometary comae, contributing to the abundance of the super-volatiles O₂ and CO. Production of O₂ from CO₂ was explicitly disregarded in the coma of comet 67P early on in the Rosetta mission,⁶ however, the situation is not as clear post-perihelion, when CO₂ can reach abundancies as high as 32% relative to H₂O. [Bockelee-Morvan et al. MNRAS 462, S170 (2016)] Even if O₂ abundance in 67P correlates better with H₂O pre-perihelion,⁶ dynamic contributions to the detected O₂ from CO₂ dissociation may still be important.”

2. Based on the statements in the paper and the references it is unclear whether the authors are proposing that comet-produced O₂ is delivered to the young Earth or O₂ produced through CO₂ collisions in Earth’s atmosphere. The former does not seem a good fit with the assumptions in 1., i.e. that the O₂ in comets are not indigenous to the comet, but rather produced through collisions in the comet coma. And even if some of the coma-produced O₂ re-adsorbed on the comet, the O₂ delivered to Earth this way would have a rather short lifetime in Earth’s atmosphere. The second interpretation is also difficult to understand, since the authors do not give any source for fast CO₂ collisions on the young Earth.

Answer: Again, we do not propose that some O₂ has been brought to Earth by comets. We rather allude to trace amounts of O₂ in the young Earth, formed in its presumed CO₂ atmosphere by bombarding objects such as micrometeorites, asteroids, etc. This is effectively the inverse of our experiment, where a target surface (meteorite, etc.) is moving against a stagnant CO₂ atmosphere with correspondingly high velocities. We rewrote the text to make that clear:

In these situations, the target surface is moving against a stagnant CO₂ atmosphere with correspondingly high velocities, driving the transformation of CO₂, and producing small amounts of O₂. Abundances in the 1000’s of parts per million measured at Mars [Hartogh et al. Astron. Astrophys. 521, L49 (2010)] may contain significant contributions from such processes.

3. Similar to the Earth-case, it is unclear whether comet-impact delivered oxygen would survive long enough to make a difference. Even transient oxygenation events may be important for remote atmosphere evaluations, but as mentioned above the proposed reaction would not in fact affect

comet bulk O₂. If the authors are instead imagining local production channels through accelerated CO₂ collisions, this needs to be made explicit and motivated.

Answer: See Point#2 above.

4. It is difficult to imagine that accelerating CO₂ to 10s of eV would be a competitive O₂ production pathway on Mars. This motivation would seem more realistic if there are reasons to think that this conversion chemistry could take place in less energetic impacts than used in this experiment.

Answer: Indeed so! We do not insinuate that accelerated CO₂ ions would make a difference on the Martian atmosphere. However, we envision our reaction taking place in a man-made plasma device with no moving parts, which may be operated on Mars with solar-generated electricity.

ⁱ. Lu, Z., Chang, Y.C., Yin, Q.Z., Ng, C.Y. & Jackson, W.M. Evidence for direct molecular oxygen production in CO₂ photodissociation. *Science* **346**, 61-64 (2014).

ⁱⁱ. Suits, A.G. & Parker, D.H. Hot molecules-off the batten path. *Science* **346**, 30 (2014).

ⁱⁱⁱ. Larimian, S. *et al.* Molecular oxygen observed by direct photoproduction from carbon dioxide. *Phys. Rev. A* **95**, 011404 (2017).

^{iv}. Wang, X.D., Gao, X.F., Xuan, C.J. & Tian, S.X. Dissociative electron attachment to CO₂ produces molecular oxygen. *Nature Chem.* **8**, 258-263 (2016).

^v. Spence, D. & Schulz, G.J. Cross sections for production of O₂⁻ and C⁻ by dissociative electron attachment in CO₂: an observation of the Renner-Teller effect. *J. Chem. Phys.* **60**, 216-220 (1974).

^{vi}. Fuserlier, S.A. *et al.* Rosina/DFMS and IES observations of 67P: ion-neutral chemistry in the coma of a weakly outgassing comet. *Astron. Astrophys.* **583**, A2 (2015).

^{vii}. Coates, A.J. Pickup particle acceleration at comets, moons and magnetospheres. *J. Phys.: Conf. Ser.* **900**, 012002 (2017).

REVIEWERS' COMMENTS:

Reviewer #1 (Remarks to the Author):

I have reviewed the author changes to my review. These changes satisfactorily address my questions and concerns. I recommend publication.

Reviewer #2 (Remarks to the Author):

The authors did mostly address the previously raised concerns, and the paper appears close to ready for publication. Two minor issues should be addressed, however:

1. The introductory sentence 'Such reactions may be important for the formation of the Solar System and Earth's prebiotic atmosphere, and for space travel to other planets' remains unnecessarily ambiguous. Even with the submitted report from the authors it is not clear what it is supposed to mean.
2. It is difficult to reconcile the low O₂ yield in the experiments with a measurable contribution of this process to the observed O₂ in the 67P comet coma. Though the authors wisely stay qualitative, the impression of the reader is that they think that their production channel may be an important contribution, while realistically it seems like it would contribute less than a % of the observed O₂. There need to be some acknowledgement of the mismatch between the O₂ yield and the O₂/CO₂ abundance ratio in the coma of 67P.

Response to Reviewer#2

Reviewer #2 (Remarks to the Author):

The authors did mostly address the previously raised concerns, and the paper appears close to ready for publication. Two minor issues should be addressed, however:

1. The introductory sentence 'Such reactions may be important for the formation of the Solar System and Earth's prebiotic atmosphere, and for space travel to other planets' remains unnecessarily ambiguous. Even with the submitted report from the authors it is not clear what it is supposed to mean.

Answer: The ambiguous sentence has been re-written, as follows: "Such reactions may offer competing explanations for the origin of O₂ in comets, in the upper atmosphere of Mars, and in Earth's prebiotic atmosphere. They may also present alternative ways for resource utilization related to space travel, such as generation of O₂ from CO₂ for making Mars habitable."

2. It is difficult to reconcile the low O₂ yield in the experiments with a measurable contribution of this process to the observed O₂ in the 67P comet coma. Though the authors wisely stay qualitative, the impression of the reader is that they think that their production channel may be an important contribution, while realistically it seems like it would contribute less than a % of the observed O₂. There need to be some acknowledgement of the mismatch between the O₂ yield and the O₂/CO₂ abundance ratio in the coma of 67P.

Answer: The results on O₂ yield are clear and the readers are free to draw their own conclusions. However, we believe that authors should also be allowed to provide their perspective. Nevertheless, we rewrote the part of the paragraph on comets to acknowledge the mismatch implied by reviewer#2. First, we point out explicitly that O₂ from CO₂ is justifiably excluded from contributing to O₂ abundance during the pre-perihelion phase. Second, we moderate the post-perihelion 10-fold increase in CO₂ abundance by stating that the level of the resulting contribution to O₂ is likely small. Third, the reviewer's request necessitates further explanation on how a low O₂ reaction yield can still be important if the reaction takes place on the surfaces of the Rosetta spacecraft. The paragraph has been rewritten as follows—with changes highlighted in blue:

"Production of O₂ from CO₂ was explicitly disregarded in the coma of comet 67P early on (pre-perihelion) during the Rosetta mission, owing to the low abundance of CO₂ and poor correlation between O₂ and CO₂ fluxes. However, the situation may warrant reexamination in the post-perihelion phase, when CO₂ can reach abundancies as high as 32% relative to H₂O, a 10-fold increase versus pre-perihelion. The precise level of contribution to the O₂ abundance in the coma cannot be determined without CO₂ ion energy and flux data. Nevertheless, the number is likely small for collisional encounters on dust and cometary surfaces. Even at low yield, however, contribution to the measured O₂ abundance may be disproportionate if the CO₂ reaction occurs close to the point of measurement. For example, we have verified experimentally that the reaction takes place on Indium-Tin Oxide (ITO), a man-made material found on Rosetta's thermal insulation and solar panels. Thus, CO₂ collisions on the spacecraft's exposed surfaces can change the composition of the surrounding gaseous halo with unknown repercussions for mass spectrometric measurements."